# Development and Evaluation of a Calibrating System for the Application Rate Control of a Seed-Fertilizer Drill Machine with Fluted Rollers

**Hongfeng Yu [1,2], Yongqian Ding [1,3],\*** , **Zhuo Liu [1], Xiuqing Fu [1,2], Xianglin Dou [1] and Chuanlei Yang [1]**

1 College of Engineering, Nanjing Agricultural University, Nanjing 210031, China; hongfengyu@njau.edu.cn (H.Y.); 2018112028@njau.edu.cn (Z.L.); fuxiuqing@njau.edu.cn (X.F.); xianglindou@njau.edu.cn (X.D.); 2018812043@njau.edu.cn (C.Y.)
2 Jiangsu Key Laboratory for Intelligent Agriculture Equipment, Nanjing 210031, China
3 National Engineering and Technology Center for Information Agriculture, Nanjing 210095, China
\* Correspondence: yongqiand@njau.edu.cn; Tel.: +86-137-7085-3275

**Abstract:** Seed-fertilizer drill machines with fluted rollers generally utilize a calibrated relationship between the fluted-roller rotation speed and mass flow rate to control the application rate. However, this relationship model gradually deteriorates with operating time and is easily affected by working conditions. To maintain the initial operating accuracy, a self-calibrating system for adjusting the control model parameters was designed. It utilizes the application-rate information and fluted-rollers rotation speed data during the operation process to dynamically establish the calibrated relationship. During the field tests performed 18 times, the self-calibrating system could dynamically re-calibrate the relationship model. After the model parameters were updated, the application rate control system maintained a high accuracy level, though the relative deviation of control parameters was relatively higher. The average relative deviations between actual and theoretical cumulative application rates were 0.97% and 1.22% for seeding and fertilizing, respectively, and the standard deviations were 0.69% and 0.62%. Correspondingly, the average relative deviations of control parameters were 7.29% and 7.86%, and the standard deviations were 1.16% and 3.06%. These results indicate that the proposed method with closed-looped control can maintain high-quality control performance and improve the productivity of the drill machine, which will benefit the agricultural production.

**Keywords:** application rate control; seed-fertilizer drill machine; fluted rollers; parameters calibration

## 1. Introduction

Precision fertilizing and seeding operations are two key parts of the wheat cultivation process. Precision control of the application rate and the wheat seed-fertilizer drill machine has been of interest to researchers [1,2]. Precision control of the wheat application rate depends not only on the measurement and control technology, but also on the performance of the seed-fertilizer drill machine [3].

Zhang et al. proposed a variable rate fertilizing technology. This control system was based on a single-chip microcomputer. The input was the forward speed of the machine and the target application rate of the prescription chart. The output was the rotation speed of the fertilizing and seeding driving shaft, which determined the actual application rate [4]. The motor is generally divided into three types: the stepper motor [5], DC(Direct-current) motor [6], and hydraulic motor [7]. According to their respective control principles, the speed control can be realized. These typical methods of controlling the application rate are, essentially, open-looped control methods. Their common characteristic is to

depend on the relationship model between the mass flow rate of fertilizing or seeding and the rotation speed of the driving shaft to the control. Based on this relationship model, real-time control of the driving shaft rotation speed can be performed to control the application rate when the target application rate and vehicle speed are given [8]. This kind of open-looped control structure is simple and economic. The disadvantages are that it cannot eliminate the error caused by interference and the control accuracy easily decreases once the relationship model parameters change because of the change of working conditions. The closed-looped control structure can effectively suppress this interference and improve the response performance of the control system [9].

Closed-loop application-rate control structure has not been widely applied, due to the lack of efficient and real-time application-rate measurement sensors. Over the past 30 years, agricultural scientists and technicians around the world have conducted extensive research on seeding and fertilizing application-rate measurement sensors. The main techniques for application rate measurement include the photoelectric method [10,11], impulse method [12,13], capacitance method [14], and weighing method [15]. The principle of the photoelectric method is that the photoelectric sensor will be blocked in the process of material flow and a photoelectric signal will be generated. The quantity of material can be calculated by the sparse degree of the photoelectric signal to measure the mass flow rate. However, the characteristics of materials (such as density, moisture content, and transparency) will have an impact on the measurement accuracy, which needs to be calibrated frequently. At the same time, the probe gets easily covered with fertilizer powder, therefore, it needs to be cleaned regularly [16,17]. The principle of the impulse method is that the material impacts the impulse sensor at a certain angle, and the measured impulse is converted into an electrical signal. The real-time application rate is then estimated indirectly by electrical signal intensity. The impulse method is mainly applied to the yield monitoring system of combine harvesters, and the yield map is obtained by combining the speed sensor and GPS (Global Positioning System) information. The impulse method is suitable for large-range mass flow rate detection [18–20]. The capacitance method measures the application rate by the difference in the dielectric coefficient between material and air. However, it requires repeated calibration due to the limitation of the material characteristics and types in actual field operation. The measurement accuracy is also difficult to guarantee [21,22]. The weighing method is used to calculate the real-time and cumulative application rate by measuring the material weight in real-time. The weighing method is simple in principle and intuitive in measurement. The difficulty lies in overcoming the vibration interference from the machine during field operation and reliably extracting the weighing signal [23]. These methods have been proven to effectively measure the cumulative application rate over a certain period, but the feasibility of closed-loop control for the application rate of wheat fertilizing and seeding operations has not been reported.

For wheat seeding and fertilizing applications, an applicator with fluted rollers is widely used in the seed-fertilizer drill machine, due to its reliability and stability, and is especially suitable for the seeding of non-single seeds or granular material, such as solid granular fertilizer and wheat [24,25]. The application rate of seeding and fertilizing applicators with fluted rollers is dependent on the material density, material filling coefficient, driving layer characteristic coefficient, wear and tear (brush damage, gap change between brush and fluted rollers), material adhesion, and clogging [26]. Further, the relationship model between the mass flow rate and the rotation speed changes over time due to these factors. Consequently, the precision of wheat seeding and the fertilizing application rate is impacted negatively if calibration is not carried out frequently. The calibration relationship model between the fluted-roller mass flow rate and rotation speed is relatively stable over a short period because there are no noticeable deformations or damage in the brush, which is installed inside the fluted-rollers. Hence, it is more economical and feasible to improve the accuracy of the application rate by calibrating the open–closed relationship between the mass flow rate and rotation speed in a timely manner, based on test results during the operation over a set period.

Due to the lack of reliable, high-precision, and real-time application rate detection methods and devices, it is difficult to establish a closed-loop control system to control the application rate

in actual production. The accuracy of the control system for controlling the application rate depends on the correctness of the control model. The control model commonly used at present is based on the calibration relationship model of the mass flow rate and the rotation speed of the fluted-roller driving shaft. The parameters of the calibration relationship model will change with the increase of working time and the change of working conditions, which affect the control performance of the control system. The aim of this paper is to establish a method which can self-calibrate the parameters of the relationship model according to the measured data while working to dynamically adjust the parameters of the control model and maintain the control accuracy of the control system that is running in a closed-loop. In this paper, the application rate measurement device [27], which was based on the weighing principle (and previously designed by the author's team), was used to measure the relationship model between the dynamic cumulative application rate and the dynamic cumulative rotation cycles in a certain period of time. Furthermore, the ratio coefficient between the mass flow rate and rotation speed can be obtained using the established relationship model.

## 2. Materials and Methods

This section mainly introduces the method of constructing a closed-looped control system which is based on the self-developed application rate measuring system and control parameter self-calibrating strategy. The commonly used form and existing problems of the control system are described in detail. We focus on the structure, control strategy, and implementation algorithms of the closed-looped control system to provide a reference for researchers with similar research interests.

### 2.1. Application Rate Control Model

The real-time fertilizing and seeding application rate of seed-fertilizer drill machine can be expressed with the following equation:

$$M(t) = \frac{c \cdot F(t)}{B \cdot V(t)}, \tag{1}$$

where $M(t)$ is the fertilizing or seeding application rate (kg/ha), $F(t)$ is the material mass flow rate (kg/min), $B$ is the operation width (m), $V(t)$ is the vehicle speed (km/h), and $c$ refers to the unit conversion coefficient between different variables. The relationship model between the rotation speed $R(t)$ of the fluted-roller driving shaft and the mass flow rate $F(t)$ of the seed-fertilizer drill machine can be established as a linear equation by calibration, and can be expressed as:

$$F(t) = k \cdot R(t) + b, \tag{2}$$

where $k$ is the ratio coefficient of the relationship model between the mass flow rate and rotation speed and represents the amount of material flowing from each single rotation of the fluted rollers (kg/r) and $b$ is the constant term of the fitting equation (kg/min).

Using Equations (1) and (2), the rotation speed of the fluted-roller driving shaft can be expressed as Equation (3).

$$R(t) = \frac{1}{k}\left(\frac{M(t) \cdot B \cdot V(t)}{c} - b\right), \tag{3}$$

Equation (3) is the basic control model for controlling the application rate of the seed-fertilizer drill machine with fluted rollers. In practice, the main controller calculates the driving shaft rotation speed in real-time, according to the relationship model shown in Equation (3), combining the target application rate with the vehicle speed to ensure that the actual application rate matches the target application rate.

## 2.2. Closed-Looped Control Strategy

In general, the application rate control was realized using the strategy described in Figure 1. It was essentially an open-looped method because there was no feedback data for updating parameters *k* and *b*. *k* and *b* were the parameters of the control model expressed as Equation (3). The value of the parameters (*k* and *b*) varied with the increase of usage time or changes in working conditions. If there was no effective device to sense the change of the model parameters, the fluted-roller rotation speed calculated from Equation (3) deviated the actual control target and caused a control error. Therefore, in order to maintain excellent control performance, effective methods were needed to sense the change of *k* and *b* and dynamically adjust the control model parameters.

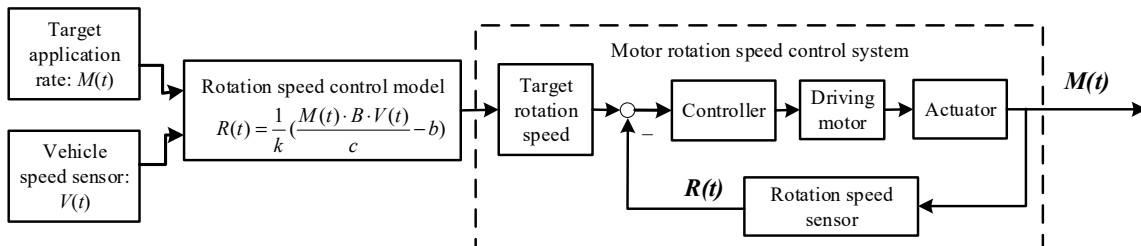

**Figure 1.** Block diagram of an open-looped control strategy for the application rate control.

A closed-looped control system for controlling the application rate was constructed, which is shown in Figure 2. The parameters of the self-calibrating system and application rate measuring system were introduced as the feedback channel to establish a closed-looped control system. The new relationship model between the mass flow rate and rotation speed was linearly fitted by recording the dynamic cumulative application rate *Q*(t) and the dynamic cumulative rotation cycles *N*(t) in a certain period. The parameters of the new relationship model were compared to the original model parameters, and then the control model parameters were corrected and updated according to the preset conditions. The *k* and *b* measured by the calibrating system replaced $k_0$ and $b_0$ in the original calibration model to maintain the control accuracy of the application rate.

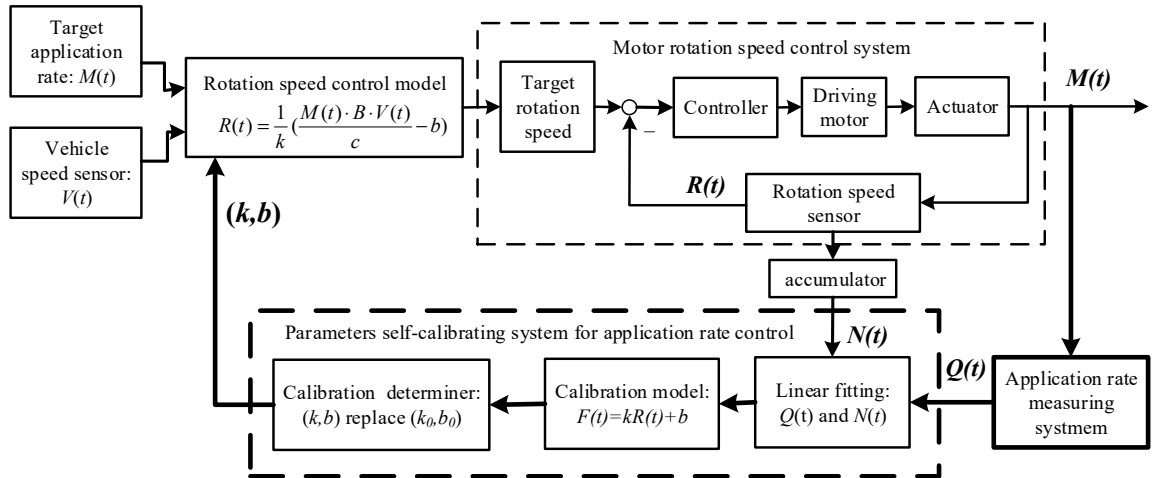

**Figure 2.** Block diagram of a closed-looped control strategy for the application rate control based on the parameter self-calibrating system.

## 2.3. Parameter Self-Calibrating Algorithm for the Application Rate Control

The flow chart of the parameter self-calibrating algorithm for application rate control is shown in Figure 3. The cumulative increment signal of the weighing sensor *D*(*t*) was obtained by the application rate measuring system, the vehicle speed *V*(*t*) was obtained by the speed measuring sensor, and

the rotation speed $R(t)$ of the fluted rollers was obtained by a rotary encoder. The control parameter self-calibrating system, combining the sensor signals from multiple sources, obtained the latest relationship model between the mass flow rate and rotation speed during the actual operation of the drill machine. The specific algorithm is described as follows.

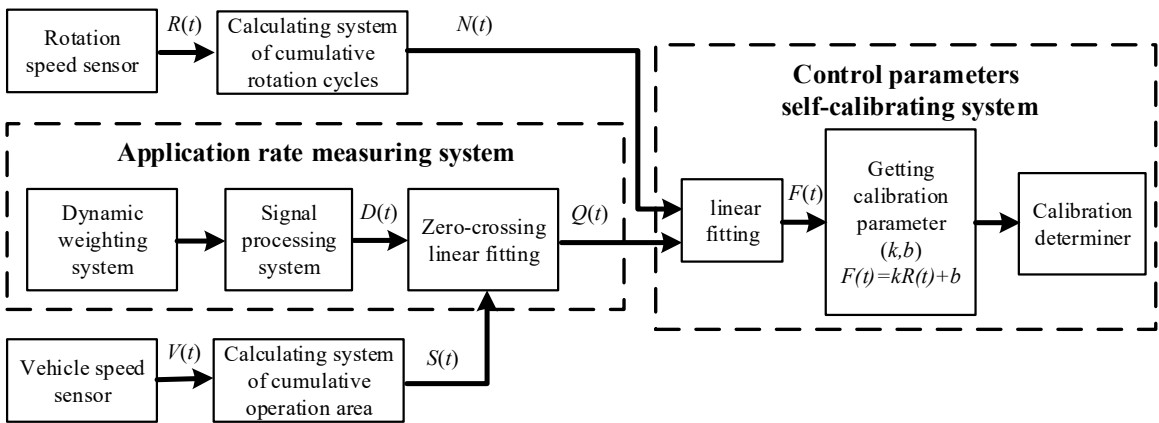

**Figure 3.** Algorithmic design of the application rate calibration system.

(1)    Acquisition of dynamic cumulative application rate Q(t)

The dynamic cumulative increment of the weighing sensor $D(t)$ was obtained by the application rate measuring system and was used to determine the cumulative application rate. However, the low stability of the $D(t)$ signal, due to the complexity of field operations, required a more complicated processing method. To obtain more accurate dynamic cumulative application rate signals, a least squares method [28,29] was used to perform a zero-crossing linear fitting of the dynamic cumulative increment $D(t)$ and dynamic cumulative operation area $S(t)$. The linear fitting results gave the dynamic cumulative application rate $Q(t)$. $S(t)$ was calculated by accumulating the product of vehicle speed and operation width of the seed-fertilizer drill machine.

(2)    Acquisition of the relationship model between mass flow rate and rotation speed

Linear fitting of the dynamic cumulative application rate data $Q(t)$ and dynamic cumulative rotation cycles $N(t)$ over a certain observation time $T$ was used to obtain the relationship model between the mass flow rate and the rotation speed during the operation of the seed-fertilizer drill machine. The parameters of the linear fitting equation included the calibration parameters $k$ and $b$, and $T$ were determined according to the actual operation condition. The dynamic cumulative rotation cycles $N(t)$ were calculated by accumulating the rotation speed of the fluted rollers over the observation time $T$.

### 2.4. Hardware System for Measuring the Application Rate

In this study, an application rate measuring device [27] based on the weighing principle (previously designed by our research team) was used as the measuring device during the operation of the seed-fertilizer drill machine. A schematic diagram of the measurement device is shown in Figure 4. This device was used to acquire the cumulative application rate information during the operation of the seed-fertilizer drill machine. Along with the rotation speed information of the fluted rollers, the relationship model between the mass flow rate and speed rotation during the operation process of the seed-fertilizer drill machine was determined, which served as the basis for calibrating the control model parameters $k$ and $b$.

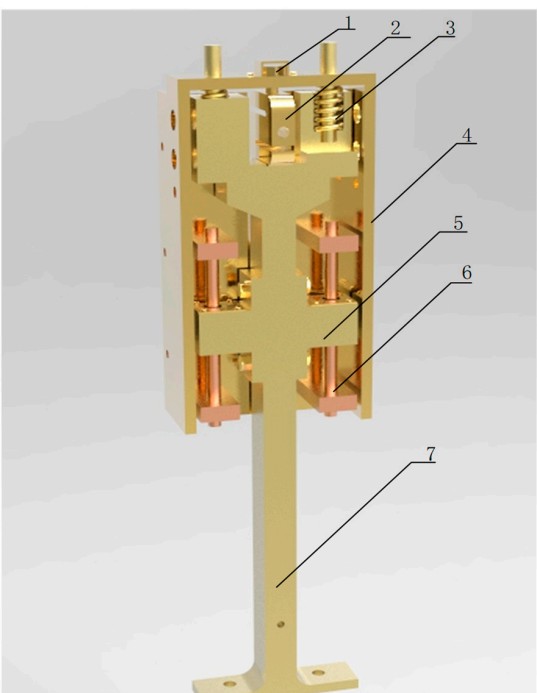

**Figure 4.** Mechanical structure diagram of the application rate measuring device. **1.** Weighting sensor connection bolts, **2.** weighing sensor, **3.** spring, **4.** box connection kits, **5.** linear bearing seat, **6.** guide shaft, and **7.** support frame.

In this paper, the S-shaped weighing sensor had a range of 100 kg and a minimum detectable mass of 0.1% FS (Full Scale) The stable support force of each spring during work was 50 kg.

## 3. Results and Discussion

The entire testing process was performed in a field which lies in the South Jiangsu Province in China. The experimental equipment with the application rate measuring and control system was installed on a rotary tiller. There were 10 seeding units and six fertilizing units installed in the seed-fertilizer drill machine. The maximum capacity of the seed tank was 80 kg, and that of the fertilizer tank was 100 kg. The operation width of the seed-fertilizer drill machine was 2.3 m, and the drill machine was attached to an 80 hp tractor. During the operation, wheat seeds and compound fertilizer were fed simultaneously to the rotary fluted rollers. To evaluate the performance, collection bins were placed underneath the seeding and fertilizing outlets. Figure 5 shows the actual experimental setup.

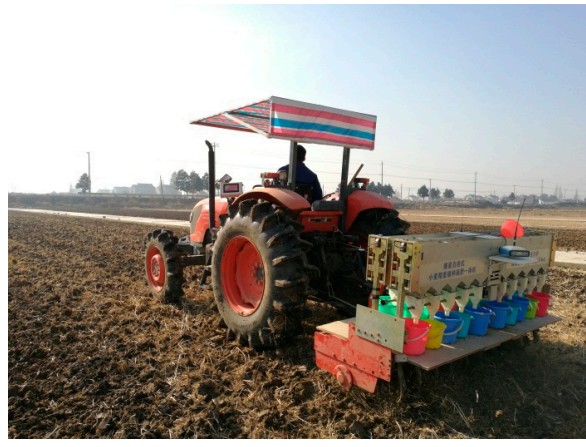

**Figure 5.** Field experiment of application rate calibration.

### 3.1. Test Conditions and Test Scheme

#### 3.1.1. Test Conditions

The entire testing process was divided into three stages from 13 to 15 December 2018. The weather condition was usually cloudy during tests. Each stage was arranged in one day to complete three groups of experiments. Each group of experiments targeted one application rate combination of wheat and compound fertilizer and was repeated once. The three target application rate combinations were as follows: 225 and 300 kg/ha, 300 and 375 kg/ha, and 375 and 525 kg/ha, respectively, as listed in Table 1. During the tests, the tractor's driving system was fixed in a certain gear, where the operation speed changed within a range of 2 km/h to 4 km/h because of the changing field resistance or different depth on the accelerator pedal during operation. After effectively traveling for 300 m, the test was deemed complete. Because the field in which we performed the test was relatively small, we needed to turn around the tractor several times to reach a 300 m testing distance. The driving speed, the seed, and the fertilizer application rates used during tests are commonly adopted operation parameters in practice in Jiangsu Province in China. The recording interval for all measurement parameters was 0.5 s during the tests.

**Table 1.** Test scheme and operation parameters.

| Target Application Rate (kg/ha) | | Vehicle Speed (km/h) | Test Distance (m) |
|---|---|---|---|
| seeding | fertilizing | | |
| 225 | 300 | [2,4] | 300 |
| 300 | 375 | [2,4] | 300 |
| 375 | 525 | [2,4] | 300 |

#### 3.1.2. Test Scheme

Three stages of the tests carried out in the field are described as follows:

The first test stage: The seed fertilizer drill machine controlled the application rate according to the calibrated relationship model between the mass flow rate and rotation speed. After each test, the application rate calibration system fitted the relationship model between the mass flow rate and rotation speed based on the latest test data but did not update the system control parameters.

The second test stage: To significantly and quickly change the calibrated model parameters, the gap between the fluted rollers and the top of the brush was artificially shortened by 1 mm by trimming the brush length of the seed-fertilizer drill machine. This would increase the mass flow rate and change the relationship model between the mass flow rate and rotation speed. The seed-fertilizer drill machine applied the original calibrated relationship model between the mass flow rate and rotation speed to complete the same test as in the first stage.

The third test stage: Using the latest test data from the application rate calibration system, the relationship model parameters of the control system were updated to control the application rate more accurately.

After each test stage, the performance of the application rate calibration system was evaluated by comparing the difference between the relationship model parameters and the relative deviation of application rates. To evaluate the control effect of the actual application rate, bins were placed under each outlet to collect the seeds or compound fertilizers. After the completion of each test, the collected materials were weighed.

### 3.2. Calibration of the Relationship between the Mass Flow Rate and Rotation Speed

The original relationship model between the mass flow rate and rotation speed was calibrated as follows:

During each calibration test process, the driving shaft rotation speed of the fluted rollers was controlled to rotate at a constant speed at different set speeds for 3 min. By weighing the actual material collected in the output bins, the relationship model between the mass flow rate and rotation speed was established, as shown in Figure 6. The ratio coefficients of seeds and compound fertilizers were 0.1015 kg per rotation and 0.162 kg per rotation, respectively.

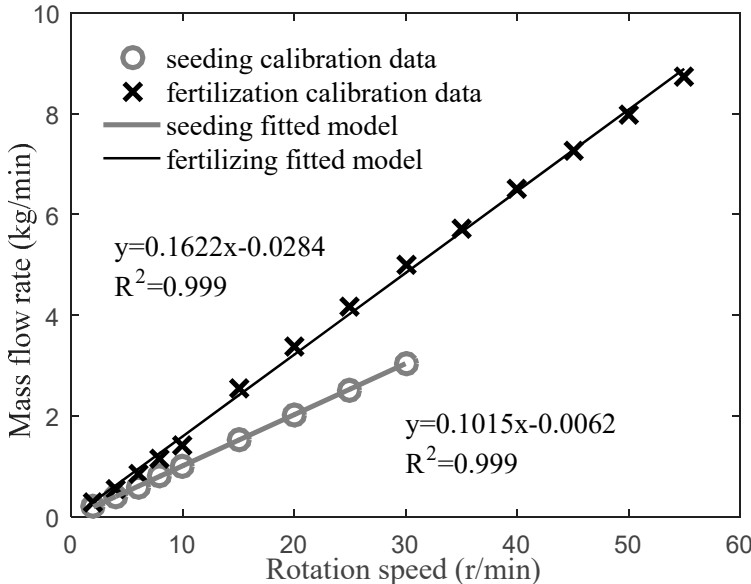

**Figure 6.** Original calibration relationship model between the mass flow rate and rotation speed.

### 3.3. Calculating Process of the Parameter Self-Calibration

To illustrate the data processing results of each step, one of the test results in the first test stage was taken as an instance in which the target application rate of seeding and fertilizing were 375 kg/ha and 525 kg/ha, respectively. First, the cumulative increment signal $D(t)$ of seeds and compound fertilizers were measured by the application rate measuring system, as shown in Figure 7. The large fluctuation in the signal was due to the lifting and lowering of the tractor when the tractor turned around.

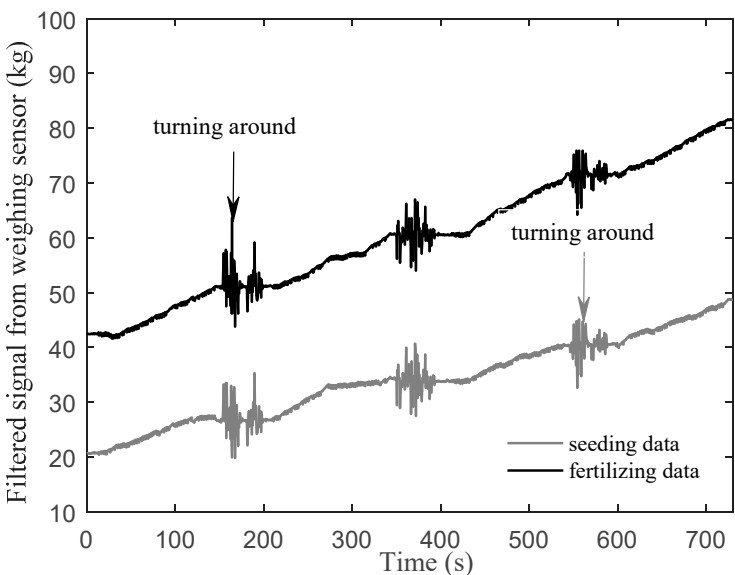

**Figure 7.** Weighing sensor signals for seeding and fertilizing.

Secondly, the zero-crossing linear fitting between $D(t)$ and $S(t)$ of the fertilizing and the seeding area was carried out, and the dynamic cumulative application rate information $Q(t)$ was calculated, as shown in Figure 8.

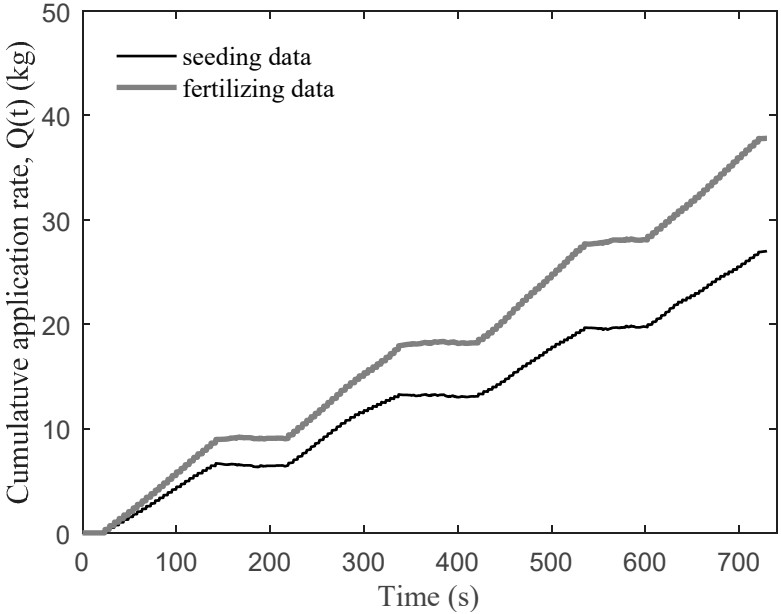

**Figure 8.** Dynamic cumulative application rate of fertilizing and seeding.

Finally, the dynamic cumulative application rate Q(t) and the dynamic cumulative rotation cycles N(t) of the fluted rollers were linearly fitted, and the relationship between the mass flow rate and rotation speed was obtained, as shown in Figure 9, from which the updated calibration model parameters (*k, b*) can be obtained.

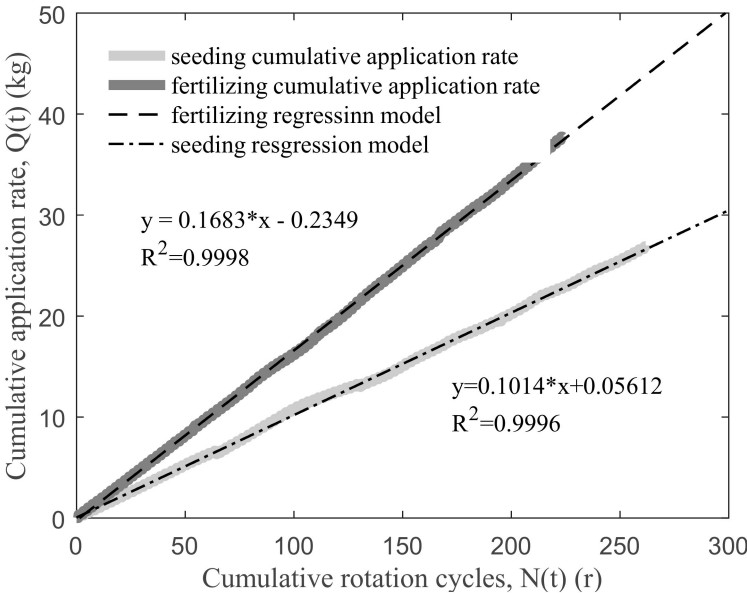

**Figure 9.** Relationship between cumulative rotation cycles and cumulative application rate.

*3.4. Test Results of the Application Rate Measuring System*

Figure 10 shows the model ratio coefficient *k* of the calibrated relationship model obtained by the application rate calibration system in the three test stages for each of the target application

rates. Figure 11 shows the absolute relative deviation after each test between the model ratio coefficient *k* measured by the application rate calibration system and the original calibration parameters. The relative deviation $E_k$ defined in Equation (4) is used to describe the change of relationship model parameters. The relevant statistical data are listed in Table 2.

$$E_k = \left| \frac{k_{real}}{k_{org}} - 1 \right| \times 100\%, \tag{4}$$

where $k_{real}$ is the model ratio coefficient measured by the application rate calibration system and $k_{org}$ is the ratio coefficient of the original relationship model.

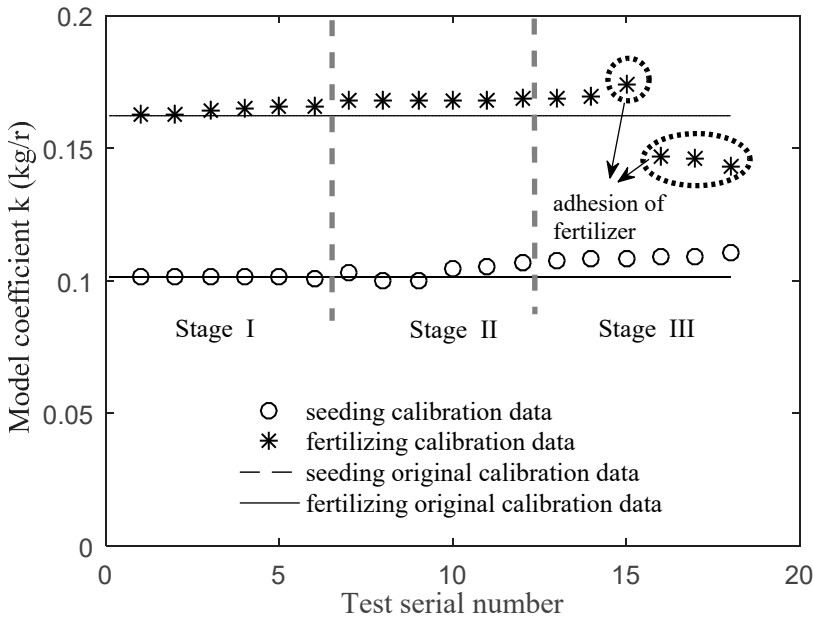

**Figure 10.** Model ratio coefficient as monitored by the application rate measuring system.

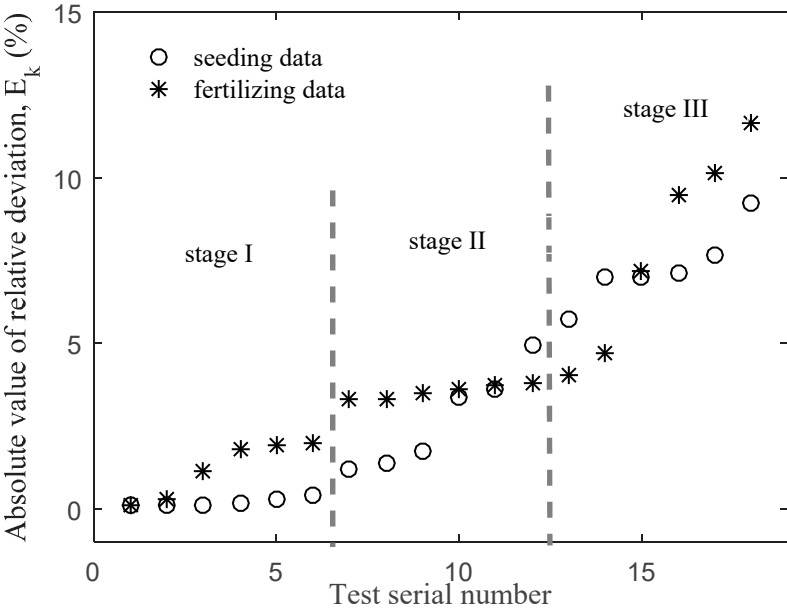

**Figure 11.** Relative deviation changing the trend of the model ratio coefficient.

**Table 2.** Statistic data of the relative deviation of model coefficient, $E_k$ (%).

|  | Stage I | | | Stage II | | | Stage III | | |
|---|---|---|---|---|---|---|---|---|---|
|  | avg | Std | max | avg | Std | max | avg | Std | max |
| Seed | 0.19 | 0.12 | 0.39 | 2.70 | 1.49 | 4.92 | 7.29 | 1.16 | 9.26 |
| Fertilizer | 1.21 | 0.82 | 1.97 | 3.56 | 0.21 | 3.82 | 7.86 | 3.06 | 11.65 |

**Note**: avg is the average relative deviation of $E_k$, Std is the standard deviation of the relative deviation of $E_k$, and max is the maximum relative deviation of $E_k$.

From the test results, it can be observed that the calibration relationship model was relatively stable for a short period. However, the model parameters for the fertilizing varied more obviously than those for seeding, probably due to the fertilizers being more prone to moisture absorption and their working performance being affected easily by environmental changes. The relative deviation of model parameters increased significantly with usage-time increase or environmental changes.

### 3.5. Test Performance of the Control Parameter Self-Calibrating System

The control model was based on the calibration relationship model between the mass flow rate and rotation speed. The calibration relationship model was established before the operation with no model parameters being updated during the process. The closer the calibration relationship model was to the actual relationship model, the higher the control accuracy. However, the actual relationship model was quite different from the original model because of the impact from working conditions. Hence, a dynamic calibration of model parameters was required during the process. In the first and second test stages, the parameters of the actual relationship model were detected based on the measured data from a certain period, but no update of the parameters was introduced. In the third test stage, the test procedure called for an update to the control model parameters to improve the control performance. To evaluate the quality of the operation of the seed-fertilizer drill machine, an evaluation indicator $E_Q$ was introduced, as shown in Equation (5).

$$E_Q = \left| \frac{W_{real}}{W_{rotation}} - 1 \right| \times 100\%, \tag{5}$$

where $W_{real}$ is the weight of materials collected by the collection container and $W_{rotation}$ is the cumulative application rate, calculated using the relationship model of the mass flow rate and rotational speed of the fluted rollers.

The evaluation indicator of each test stage is shown in Figure 12, and related statistical results are listed in Table 3.

**Table 3.** Statistical data of the relative deviation of cumulative application rate, $E_Q$ (%).

|  | Stage I | | | Stage II | | | Stage III | | |
|---|---|---|---|---|---|---|---|---|---|
|  | avg | Std | max | avg | Std | max | avg | Std | max |
| Seed | 0.59 | 0.34 | 1.00 | 3.39 | 1.16 | 4.73 | 0.97 | 0.69 | 1.61 |
| fertilizer | 1.94 | 0.62 | 2.68 | 4.88 | 0.39 | 5.25 | 1.22 * | 0.62 * | 2.00 * |

**Note**: The results with * in the table are the calculation results after removing the singular point value, and if not, the corresponding results are 1.86, 1.50, and 5.00 from left to right in the table.

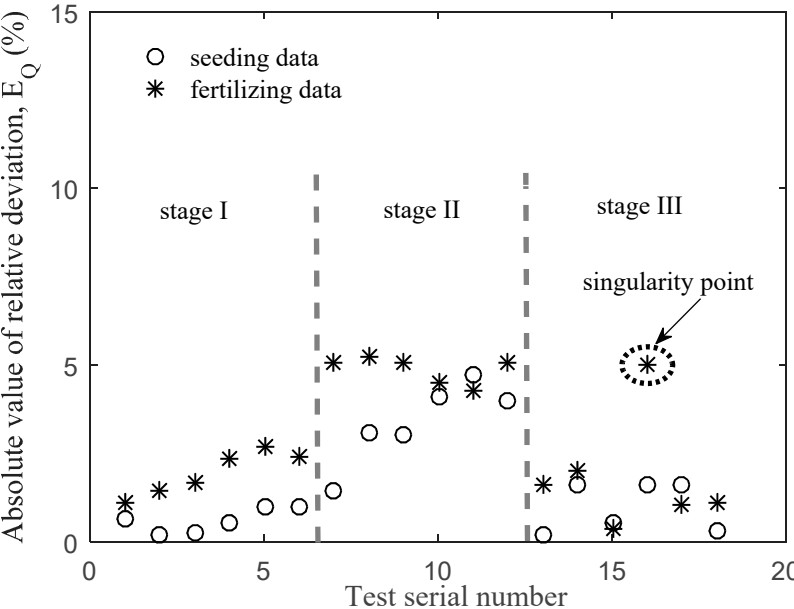

**Figure 12.** Relative deviation of the cumulative application rate in different test stages.

### 3.6. Discussion

From the test results, it can be observed that in the first test stage, the actual model ratio coefficient *k* changed slightly from its original level. Further, the relative deviation of the cumulative application rate was less than 3%, which indicated that the overall control accuracy was excellent, and the calibration relationship model was relatively stable for a short period (in one day of work in our tests). This relative stability is the main reason for the ongoing wide usage of the current open-looped model, as users tend to ignore the fact that the control error gradually increases with the operating time.

In the second and third test stages, the change of the ratio coefficient *k* was much larger than in the first test stage, as the original parameters of relationship model had been deliberately changed and the relative deviation of the ratio coefficient exhibited a gradual upward trend with the operating time. Since the control parameters were not updated in the second test stage, the relative deviation of the cumulative application rate was significantly higher than in the first test stage. This indicated that the preset control model was quite different from the actual control object model. Because of the essential open-loop of the control system, the control system could not correct the control deviation itself. With the increase of working time, the control effect of the control system weakened or even lost effect.

In the third test stage, the relative deviation of the ratio coefficient *k* further increased. Deformation or damage in the brushes appeared relatively more frequently than in the first two test stages. Further, the increased moisture absorption by the fertilizer resulted in partial caking and bonding in the rollers and brushes. These phenomena can potentially be the reason for the deviation from the original calibration relationship model. However, the relative deviation of the cumulative application rate did not increase with the increase in the ratio coefficient *k*. Conversely, it decreased to a relatively lower level, similar to the initial test stage. This is because the parameters of the self-calibrating system was in action and the closed-looped control effectively counteracted the negative effects of the parameter change on the control object. This indicates that the dynamic calibration on the parameters of the control model had been realized.

The overall test results showed that the calibration relationship model had a relatively higher stability for a short period, but the parameters of the calibration relationship model gradually deteriorated with the operating time. The relationship model needs an appropriate way to calibrate to support the control model in practice to continue with better performance. Our parameters self-calibrating method provides a feasible way to solve this problem.

## 4. Conclusions

Seed-fertilizer drill machines with fluted rollers generally use a calibration relationship model between the mass flow rate and driving shaft rotation speed before initiation to control the application rate of seeding and fertilizing. This calibration relationship model is relatively stable for a short period, but the deviation between the calibrated relationship model and the actual model gradually increases with the operating time of the drill machine because there can be significant changes (such as deformation, damage, caking or bonding by wetted fertilizer, etc.) appearing in fluted rollers, brushes, and other working parts. Thus, the accuracy of the seed and fertilizer application rate is negatively affected. This kind of control mode is essentially open-looped. To keep a high-quality operation performance, frequent calibration of the control model is inevitable. This not only affects the work efficiency, but is also not suitable for the use of ordinary farmers, since special skill is required for calibration.

In this paper, based on a self-developed application rate measuring system, we provided a method for the self-calibrating relationship model between the mass flow rate and driving shaft rotation speed. Because of the function of self-calibrating the control parameters, the control mode is closed-looped. Field tests showed that the calibrating system can effectively correct the control parameters based on the detected data from the application rate measuring system from a certain period. It can be used to hold a continuous high-quality control performance and improve actual productivity. This method can be applied to the existing mainstream seed-fertilizer drill machines, which will benefit agricultural production.

In contrast to other closed-looped control methods, we do not use the real-time feedback information of the application rate for controlling, and instead use the integrated information from a certain period to draw the control parameters. Compared to the real-time detection method, our method provides an easier way to obtain effective information in the complicated field operation environment.

It is also noted that actual field operation is complex and there are still many difficulties that need to be solved. It is difficult to determine suitable duration to calibrate new control parameters. Further, the application rate measuring system can be mixed with significant vibration interference. The proposed application rate calibration strategy is not smart enough and requires further detailed study beyond the conducted extent described in this paper.

**Author Contributions:** Conceptualization, H.Y. and Y.D.; methodology, H.Y. and Y.D.; software, Z.L. and C.Y.; validation, H.Y. and X.F.; formal analysis, X.D.; data curation, H.Y.; writing—original draft preparation, H.Y.

**Funding:** This work was mainly supported by the National Key Research and Development Program of China (Grant No. 2016YFD070030403) and partially supported by the Fundamental Research Funds for the Central Universities (Grant No. KYGX201703) and by the National Engineering and Technology Center for Information Agriculture in Nanjing Agricultural University, China.

**Acknowledgments:** We would like to thank Editage (www.editage.cn) for English language editing.

**Conflicts of Interest:** The authors declare no conflict of interest.

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
