# Peer review of "Development and Evaluation of a Calibrating System for the Application Rate Control of a Seed-Fertilizer Drill Machine with Fluted Rollers"

_applsci, doi:10.3390/app9245434_

Round 1

Reviewer 1 Report

The manuscript presents the results of a study investigating a calibrating system for the precision control of wheat seed and compound fertilizer application rates. This is an interesting and a topical issue because nowadays machine parameters are increasingly often controlled during farming operations. Below are my detailed comments that should be addressed by the Authors during the revision process to increase the scientific merit of the study and improve the clarity of presentation.

Comment 1

I am not sure whether the manuscript’s title adequately reflects its content. The first word is “design” but the design process (assumptions, stages, etc.) is not described in the main text – instead, the Authors refer to a previous article (item No. 14 in the Reference section). The scope of the present study is limited to the testing of the application rate control system. Therefore, the title could be modified as follows: “Development and evaluation of a calibrating system for the application rate control of a seed-fertilizer drill machine with fluted rollers”, or “Evaluation of a calibrating system for the application rate control of a seed-fertilizer drill machine with fluted rollers”.

Comment 2

The existing methods of application rate control (open-loop systems) have been insufficiently described in the Introduction section. The Authors have also stated that closed-loop application rate control systems have not been widely used due to the lack of efficient sensors. The disadvantages and limitations of the methods applied to date should also be briefly summarized, followed by the advantages of the methods involving the closed-loop system – such information would enable the readers to understand the rationale (justification) for your research.

Comment 3

Page 1, line 44 – insert a space between the word “method” and the reference number in brackets. The same applies to page 6, line 18.

Comment 4

In the first chapter, the Authors have mentioned a relationship between the roller rotation speed and flow rate, which is also referred to in subsequent sections. However, I am not sure if this is really a flow rate or rather a seeding rate, since seeding units are at the same time seed metering units. Flow rate is defined as the flow of mass (volume) though surface S (e.g. cross-section) per unit of time. Therefore, if the Authors use the term “flow rate”, they should also give the cross-sectional area of the outlet.

Comment 5

Page 2, line 17 - “This is because the calibration relationship between fluted-roller flow rate and rotation speed is stable over a short period” – what is the reason for that? – please provide a brief explanation.

Comment 6

Chapter 2, “Materials and Methods”, Equation (3) – in the numerator, the operation width is expressed as a function of time, which is incorrect because the operation width did not change over time. The symbols defined under the equation should be italicized, as in the equation (here and throughout the manuscript).

Comment 7

The aim of the study should be clearly stated, and the research problem should be formulated and justified.

Comment 8

The research methods should be described in greater detail. The information provided is too laconic: for instance, the vehicle speed was 2-4 km/h – were only these two speeds tested (2 and 4 km/h) or was the speed increased gradually/changed at intervals? Was the accuracy of measurements affected by the vehicle speed? – the relevant information should be provided in the Results section. Also, please note that the seed and fertilizer application rates used in practice are considerably higher than those tested in the study (?). Were working runs in the field carried out at each stage of the experiment?

Comment 9

The working parameters of the seed drill (e.g. of the seeding unit) should be provided, including e.g. the seeding width. The dimensions and working parameters of seeding units should be shown in figure drawings. The characteristics of sensors should also be provided, including their models/manufacturers, measuring range, measurement error, etc. The same applies to the weighing sensors - what was the measurement error and data recording interval for the mass, rotational speed, vehicle speed, etc.?

Comment 10

Subsection 2.2.2 – please define the term “cumulative operating width of the seed-fertilizer drill”.

Comment 11

According to the Authors, the test was deemed complete after travelling 300 m; in Figure 7, fluctuations (interference) in the filtered signal from the weighing sensor, resulting from the lifting and lowering of the tractor, are recorded every 180-200 seconds, which at the adopted vehicle speed (2-4 km/h) does not correspond to travelling the distance of 300 m.

Comment 12

Please correct the scale in the Figures, e.g. Figure 7 – up to 100 on the Y-axis, Figure 8 – up to 50, etc. Figure 9 – the first value on the Y-axis is -10. How is that possible if the application rate cannot be negative?

Comment 13

The Authors claim that each stage of the experiment was completed in one day, therefore the information given in subsection 3.4 is surprising - the model parameters for fertilizer application vary significantly, compared with seed application, due to the fact that fertilizers are more prone to moisture absorption. Did the weather conditions change so much during the day? In the description of test conditions, there is no information on weather conditions during the experiment. Therefore, please reconsider the cause underlying the observed differences.

Comment 14

What would be the calibration error if seeds of other cereal species were sown and different types of fertilizers were used?

Comment 15

The manuscript lacks an in-depth discussion of the results; the research findings were not compared with the results of studies investigating open-loop systems.

Comment 16

According to the Authors, the calibration system error increases with operating time. What is the reason for this increase? – please provide an explanation.

Reviewer 2 Report

This study considers the main objective as, "Design and Testing of Calibrating System for Application Rate Control of Seed-Fertilizer Drill Machine with Fluted Rollers". This research has a potential to contribute the current idea existing. However, this research is necessary to fix many flaws inside of the manuscript.

There are too many incorrect language details in your manuscript (lexical, grammatical and spelling errors, and phrases that do not belong to correct English). Please seek professional language assistance to ensure that language errors are eliminated and the style of writing becomes more reader-friendly. Current format the authors followed and wrote is not correct to be published that needs to follow the journal guidelines. Introduction should be rewritten. It is too short and is also not clearly stated research questions and targets. Literature review should be included and needs to be describing on what basis literature review was done? Particularly, this section lacks of enough references to support the authors objective. In the section of materials and methods, it is with the lack of a proper order in the content, it is very difficult to follow. Authors have to rethink a different flow. The discussion on results is poorly presented. Please revise your conclusion part into more details. Also, every research is bound to have some limitation. References are not followed by the journal guidelines. Please revise them carefully. Also, the quantity of references are not enough to support the idea and narration of the authors. No resources from the applied sciences. Tables and Figures are not followed by the journal guidelines that are not clear at all to read the information inside. That will fail to give a clear image for the general journal readers. Also, please reduce the quantity of figures that are redundant with the text proposed.

Reviewer 3 Report

Authors of the paper Design and Testing of Calibrating System for Application Rate Control of Seed-Fertilizer Drill Machine with Fluted Rollers presents a relevant topic considering the control of the application rate and the strategy of self-calibration of the parameters.

The existing concepts in the specialized literature and the bibliographic sources are presented limited in the paper, which is why we recommend the authors to present in a new separate chapter their "literature review".

The research methodology is relevant, and the presented research methods (example: Method of controlling the speed of application of the seed fertilizer drill, Algorithmic design of the application speed calibration system, etc.), determine us to appreciate the structuring method. of the methodology to confirm the research results.

The results of the research are adequate and presented by the research authors with practical arguments, especially due to the fact that the entire testing process as mentioned by the research authors was carried out in a field. Moreover, we consider that due to the fact that the experimental equipment’s had a measuring and control system of the application speed and that it was installed on a rotary mower, we determine to appreciate the work from the point of view of the practical utility.
Regarding the conclusions we consider that they should be reformulated both from the point of view of scientific results (the authors' contribution to scientific discoveries), but also from the point of view of the practical utility for the beneficiaries of farmers, motivating the fact that "the device for measuring the rate of self-application" developed, it is given by a self-calibration system for updating the speed control parameters in real time”.

After reviewing the paper by the research authors according to our recommendations, and as a result of the usefulness of this paper for optimizing the calibration process in the technological process at the farm level, we propose to publish the paper.

Round 2

Reviewer 1 Report

Dear authors

Uważam, że rękopis przesłany do recenzji może zostać opublikowany w obecnej formie.

Reviewer 2 Report

First of all, I appreciate to the authors for making efforts to carry out the changes by the referees. However, there are still flaws in the manuscript revised. First, it is necessary to format the manuscript following by the journal guidelines (esp. line spacing, punctuation, references including DOI, line 169, [15-16] is not correct, etc.). Second, in the last part of Introduction, it is necessary to put the objectives of this research. Third, between each section and sub-section, it is required to put some contents, i.e. 2 and 2.1. Fourth, there are still many figures that are overlapping the text so it is necessary to remove either the figure or the corresponding text. Fifth, although the authors are adding the discussion section, but there are no implications and contexualization related with the objectives the authors raised. Sixth, language is still rough and is not appropriate for academic work that is recommended to get a professional editing service.

Reviewer 3 Report

We congratulate the team of authors for the revision of the work according to the recommendations and we propose to publish the work.

Round 3

Reviewer 2 Report

First of all, I appreciate to the authors for making efforts to carry out the changes by the referees. I think the authors did a good job in clarifying the queries that this manuscript is substantially improved.